# Osteoclast Multinucleation: Review of Current Literature

**DOI:** 10.3390/ijms21165685

**Published:** 2020-08-08

**Authors:** Joe Kodama, Takashi Kaito

**Affiliations:** Department of Orthopedic Surgery, Osaka University Graduate School of Medicine, 2-2 Yamadaoka, Suita, Osaka 565-0871, Japan; joekodama@gmail.com

**Keywords:** osteoclast multinucleation, DC-STAMP, OC-STAMP, OSCAR, FcRγ, Siglec-15, DAP12, NETRIN-1, Flrt2, Unc5b

## Abstract

Multinucleation is a hallmark of osteoclast maturation. The unique and dynamic multinucleation process not only increases cell size but causes functional alterations through reconstruction of the cytoskeleton, creating the actin ring and ruffled border that enable bone resorption. Our understanding of the molecular mechanisms underlying osteoclast multinucleation has advanced considerably in this century, especially since the identification of DC-STAMP and OC-STAMP as “master fusogens”. Regarding the molecules and pathways surrounding these STAMPs, however, only limited progress has been made due to the absence of their ligands. Various molecules and mechanisms other than the STAMPs are involved in osteoclast multinucleation. In addition, several preclinical studies have explored chemicals that may be able to target osteoclast multinucleation, which could enable us to control pathogenic bone metabolism more precisely. In this review, we will focus on recent discoveries regarding the STAMPs and other molecules involved in osteoclast multinucleation.

## 1. Introduction

Osteoclasts are multinucleated cells derived from hemopoietic progenitors of the monocyte-macrophage lineage. They differentiate upon exposure to macrophage colony stimulating factor (M-CSF) and receptor activator of NF-κB ligand (RANKL), which are presented by osteoblasts and osteocytes [1]. During the osteoclast differentiation process, pre-osteoclasts first differentiate into TRAP-positive mononuclear cells, then become giant, multinucleated cells through cell-cell fusion [2] and incomplete cytokinesis [3]. Mature osteoclasts, which are giant, polarized and multinucleated, can degrade (resorb) the skeletal matrix by generating a large sealing zone which consists of an actin ring encompassing a ruffled border [4,5,6].

An exquisite balance between bone resorption by osteoclasts and bone formation by osteoblasts, known as bone remodeling, is required for multiple physiological processes such as bone growth, fracture healing and calcium-phosphorus metabolism. Any malfunction of the osteoclastic activity disharmonizes this balance. Dysfunction of osteoclasts causes osteopetrosis [7], while accelerated osteoclastic bone resorption contributes to pathological conditions including postmenopausal osteoporosis and inflammatory arthritis [8,9]. At present, bisphosphonates and anti-RANKL antibody are the first-line therapies targeting excessive osteoclast activity, but their general suppression of osteoclasts inevitably affects bone formation due to the delicate balance between osteoclast and osteoblast activity [10,11]. 

As a hallmark of osteoclast maturation, multinucleation occurs at a late phase of osteoclast differentiation. Multiple studies have demonstrated that osteoclasts that fail to multinucleate nevertheless keep their osteoclast phenotype, express osteoclast-related markers such as TRAP and Cathepsin K, and retain low levels of bone resorption ability [12,13,14,15,16,17]. This suggests that therapies targeting osteoclast multinucleation may reduce rather than eliminate their bone resorption activity, thereby allowing us to control bone destruction more selectively and circumventing the negative effects of general osteoblast suppression. To this end, many researchers have explored the molecular mechanisms of osteoclast multinucleation. The identification of DC-STAMP and OC-STAMP, two “master fusogens” in osteoclasts, was a breakthrough in the first decade of the 21st century [16,17]. As the direct ligands for these two cell-membrane receptors have not been identified, however, the investigation of their downstream signaling pathways has been difficult. Despite the identification of several other molecules regulating osteoclast multinucleation, the overall view of the multinucleation mechanism remains obscure.

In this review, we first recount the identified molecules that activate DC-STAMP or OC-STAMP or are activated by them. Next, we will outline the molecules and pathways that are not directly related to these two STAMPs. Finally, we will discuss recent preclinical studies focusing on osteoclast multinucleation.

## 2. DC-STAMP

DC-STAMP was first identified through a random cDNA library sequencing of human monocyte-derived dendritic cells. This 470-amino-acid protein is expressed at the cell surface [18] and on the endoplasmic reticulum (ER) membrane [19,20], and contains seven putative transmembrane regions with an intracellular C terminus. This molecule was discovered to be involved in bone metabolism through a cDNA subtraction screening between multinucleated osteoclasts and mononuclear macrophages [21]. Contrary to Kukita’s conjecture that DC-STAMP was involved in osteoclast differentiation, Yagi et al. used DC-STAMP-deficient mice to prove that DC-STAMP is a key regulator in cell-cell fusion of osteoclasts and foreign body giant cells (FBGCs) [17]. TRAP-positive mononuclear osteoclasts were observed in DC-STAMP-/- mice, and their expression levels of osteoclast markers and transcription factors such as c-Fos, NFATc1 and Cathepsin-K were comparable to those in wild-type mice, yet their osteoclast multinucleation was completely abrogated. DC-STAMP-overexpressing transgenic mice, on the other hand, exhibited an osteoporotic phenotype because of hyper-multinucleation of osteoclasts. It is worth noting that, although DC-STAMP expression was driven by an actin promoter, over-stimulation was not observed in other cell types such as myotubes and germ cells, suggesting that DC-STAMP regulation is tissue-specific [22]. 

### 2.1. DC-STAMP Is a Transmembrane Protein

DC-STAMP is encoded by the gene *Tm7sf4* located on chromosome 8 in humans (locus 8q22.3). *Tm7sf4* is a single, widely conserved gene in mammals. It contains four exons and three introns, in which seven putative transmembrane regions and an intracellular C-terminus are located between exon 2 and exon 4 [23,24]. Hartgers et al. have speculated that intron 1 in the 5′ untranslated regions might possess binding sites for transcription factors [23].

Human DC-STAMP consists of 470 amino acids and has a molecular mass of about 53 kDa [18]. Although it is widely accepted that DC-STAMP is a multiple transmembrane (TM) protein, it remains unknown how many times it passes through the cytoplasmic membrane. In Yagi’s study, hydrophobicity analysis showed seven putative TM regions, but TM2, TM3 and TM4 were relatively weak. They proposed a seven-transmembrane model in which an uncleavable, hydrophobic sequence of amino acids at the N terminus extends out from the cell membrane and a C terminus possessing several phosphorylation sites is located intracellularly for downstream signaling [18]. In contrast, Yang et al. have predicted a six-transmembrane topology based on a TMHMM (Transmembrane Hidden Markov Model) algorithm for DC-STAMP and OC-STAMP [16]. Witwicka et al. provided further evidence that OC-STAMP has a six-transmembrane structure with both terminuses located intracellularly and suggested the same topology for DC-STAMP [14]. Further investigation is needed to answer this question.

The cytoplasmic tail of DC-STAMP contains several serine residues, two of which are considered targets for phosphorylation. The details of the intracellular reactions are still largely unknown, however.

### 2.2. DC-STAMP Works in a Receptor-Ligand Machinery

There is widespread agreement that DC-STAMP works through receptor-ligand interaction. The unique structure of *DC-STAMP* enables an intron at the 5′ untranslated region to imitate several transmembrane receptors such as prostaglandin EP4 receptor gene [25] and human CC chemokine receptor 5 (CCR5) gene [26]. Yagi et al. visually demonstrated the receptor-ligand interaction model of DC-STAMP: EGFP-positive DC-STAMP-/- cells fused with wild-type osteoclast precursors, suggesting that DC-STAMP expression on the cell membrane is indispensable only on one side of the two fusing cells and that the fusing partner might present a ligand for DC-STAMP. The existence of a soluble ligand was denied based on the finding that supernatants of DC-STAMP-/- osteoclast cultures did not induce fusion in wide-type osteoclasts and vice versa [17]. Because osteoclast precursors without DC-STAMP expression do not fuse by themselves [17], the putative mechanism is that a membrane-bound ligand on the fusion donor cell ligates to DC-STAMP on the fusion founder cell to initiate the fusion process.

To date, the identity of DC-STAMP’s ligand remains unclear. Due to the structural similarity between DC-STAMP and chemokine receptors, all of which are seven-transmembrane receptors, Yagi et al. suggested that the potential ligand of DC-STAMP might be a chemokine. The chemokine (C-C motif) ligand 2 (CCL2), also known as monocyte chemoattractant protein-1 (MCP-1), which is induced by RANKL and which promotes osteoclast fusion in humans [27], is one possible ligand. Recently, several proteins have been found to bind directly with DC-STAMP, such as CCN family 2/Connective tissue growth factor [28], OS9 [29] and Luman [30]. These might be ligands of DC-STAMP as well. The macrophage-lineage specific fusogen OC-STAMP has also been shown to interact with DC-STAMP and is also dispensable for osteoclast fusion, making it a candidate for the ligand as well [15]. Identifying the ligand of DC-STAMP is an urgent issue as it will enable us to explore its downstream signals.

### 2.3. The Expression and Cellular Distribution of DC-STAMP Change under RANKL Signaling

There is some debate about whether DC-STAMP in osteoclasts is initially induced by RANKL. Kukita et al. were the first to identify the protein of DC-STAMP expressed in osteoclasts. In their experiment, murine RAW264.7 cells did not express DC-STAMP mRNA without RANKL stimulation [21]. This phenomenon was further confirmed in mouse bone marrow mononuclear cells and macrophage-like cell-lines [17,22,31]. On the other hand, Yagi et al.’s work in 2007 using RT-PCR showed that RAW264.7 cells expressed a certain level of DC-STAMP in the absence of RANKL stimulation, although this level was upregulated under RANKL treatment [32]. In the same research, spleen mononuclear cells isolated from wild-type mice expressed DC-STAMP only after RANKL-stimulation. In human cells, Hartgers showed that mRNA of DC-STAMP was only expressed in dendritic cells, not in undifferentiated monocytes elutriated from peripheral blood [18]. At the protein level, however, Mensah’s group used a novel monoclonal antibody to detect DC-STAMP expression on cell surfaces through flow cytometry and demonstrated that DC-STAMP was expressed on the surfaces of both RAW264.7 cells and primary mouse bone marrow mononuclear cells without RANKL induction. They suggested that RANKL stimulation causes the internalization of surface DC-STAMP, and that the DC-STAMP mRNA upregulation induced by RANKL sustains the expression of DC-STAMP as the membrane receptor [33]. Another intriguing finding is that, after three days of maturation under RANKL stimulation, both mRNA and protein level of DC-STAMP in multinuclear osteoclasts decreased dramatically, while expression levels of other osteoclast marker genes and fusion-related genes increased. Chiu et al. confirmed the existence of surface DC-STAMP in freshly isolated monocytes purified from human peripheral blood mononuclear cells, whereas its expression level was downregulated in mature osteoclasts [34]. By overexpressing DC-STAMP in RAW264.7 cells, they further proved that continuous upregulation of DC-STAMP might inhibit the differentiation of RAW264.7 cells into mature osteoclasts [34]. Taking these findings together, we can draw some preliminary conclusions about the molecular dynamics of DC-STAMP during osteoclast multinucleation: (1) osteoclast precursors may possess this receptor on their cell surfaces; (2) in the early stages of differentiation, surface DC-STAMP may be internalized while its mRNA expression is induced, promoting osteoclast fusion; and (3) both mRNA and protein level may decline as multinucleation progresses to pave the way for the late phase of osteoclastogenesis.

### 2.4. C-Fos and NFATc1 Modulate the Gene Expression of DC-STAMP

RANKL signaling induces the expression of c-Fos, a component of activator protein-1 (AP-1) positioned upstream of NFATc1. AP-1 binds to NFATc1’s promoter region to induce the latter’s expression [35]. The expression of NFATc1 is also rapidly enhanced by its own auto-amplification [36], triggering the onset of osteoclastogenesis. These two master transcriptional factors induce several molecules that are essential for osteoclastogenesis [36,37], including DC-STAMP.

Yagi et al. identified two tandem binding sites for AP-1 and NFATc1 in the promotor region of DC-STAMP whose mutation abolished DC-STAMP expression [32]. They also demonstrated that AP-1 and NFATc1 are essential for osteoclast fusion but are dispensable for foreign body giant cell fusion, which indicates that this pathway is cell-type-specific. Kim et al. confirmed that NFATc1 inactivation attenuates expression of DC-STAMP and ATP6v0d2 as well as osteoclast fusion, while overexpression of DC-STAMP and ATP6v0d2 rescues the formation of multinucleated osteoclasts [31].

There might be a reciprocal regulation between DC-STAMP and NFATc1. An immunoreceptor tyrosine-based inhibitory motif (ITIM) has been identified in the intracellular tail of DC-STAMP [34]. Recently, the same group made a new breakthrough by creating a light-activatable DC-STAMP chimeric molecule to mimic its ligation [13]. They found that DC-STAMP’s downstream signaling via ITIM is essential for cell-cell fusion. In addition, ITIM-deleted DC-STAMP impaired the nuclear translocation of NFATc1, while knockout of DC-STAMP downregulated the expressions of NFATc1 at both gene and protein level [13], suggesting that DC-STAMP also regulates NFATc1.

### 2.5. Extracellular Signal-Regulated Kinases (ERKs) Regulate DC-STAMP

ERK1 and ERK2 belong to the MAPK family and are located downstream of the Ras-Raf-MEK cascade, also known as the canonical MAPK signaling pathway. The widely expressed ERK1 and ERK2 are activated mainly by growth factors and regulate proliferation, survival, differentiation and protein synthesis in several cell lineages [38]. 

He et al. used conditional KO mouse of ERK1 and ERK2 to investigate their roles in osteoclasts [39]. They demonstrated that ERK1 positively regulated osteoclastogenesis, and that its disruption decreased multinucleated osteoclast formation from bone marrow mononuclear cells. A later report by Oh et al. demonstrated that blockage of ERK1/2 by an ERK inhibitor significantly inhibited the expression of DC-STAMP and Atp6v0s2, and completely eliminated the formation of large, multinucleated osteoclasts [40]. In addition, two studies have shown the reciprocal regulation of endogenous and exogenous collagenases on DC-STAMP through the MAPK cascade [41,42]. Exogenous MMP-13 induced the Raf-MEK-ERK1/2 cascade, activated the NFATc1-DC-STAMP axis, and subsequently enhanced osteoclast multinucleation [41]. In contrast, inhibition of endogenous MMP-8 and MMP-13 activated ERK1/2, especially ERK2, and enhanced the expression of DC-STAMP and osteoclast fusion, which is independent of the osteoclastogenesis initiated by NFATc1 and NF-kB [42]. Taken together, these findings suggest that, regardless of the involvement of the NFATc1 or NF-kB axis, DC-STAMP seems to be regulated by the canonical MAPK cascade involving ERKs.

### 2.6. Bcl6 Is a Transcriptional Repressor of DC-STAMP

B lymphocyte-induced maturation protein 1 (Blimp1) and B cell lymphoma 6 (Bcl6) are two transcriptional repressors involved in osteoclastogenesis. Bcl6 is an antagonist of NFATc1 that binds to the promoter regions and suppress the expressions of osteoclastic genes such as *Nfatc1*, *Ctsk* and *Dc-stamp*, all of which are targeted by NFATc1 [32,36,43]. Blimp1, on the other hand, is a RANKL-induced transcriptional repressor of Bcl6. RANKL induces the expression of Blimp1 to suppress Bcl6 so that NFATc1 can bind to osteoclastic genes, including DC-STAMP [44]. Ihn et al. administered a derivative of imidazole (KP-A038) to murine bone marrow cells, which decreased Blimp1, increased Bcl6 and subsequently suppressed the expressions of NFATc1 and DC-STAMP [45]. A very recent study demonstrated that treatment with connective tissue growth factor (CTGF), the second member of the CCN family of proteins (CCN2), specifically upregulated the expressions of DC-STAMP and ATP6v0d2, facilitating the fusion of mouse bone marrow-derived macrophages into multinucleated osteoclasts by downregulating the expression of Bcl6 without affecting any other pro-osteoclastogenesis factors [46]. This result further underlines the significance of the signaling between Bcl6 and DC-STAMP for osteoclast multinucleation. 

### 2.7. MITF-PU.1 Complex and Tal1 Work Reciprocally to Regulate DC-STAMP Expression

The combination of microphthalmia-associated transcription factor (MITF), a basic-helix-loop-helix zipper protein, and the ETS (E26 transformation-specific) family transcription factor PU.1 has been demonstrated to activate several osteoclastic genes such as *Trap*, *Oscar* and *Dc-stamp* [47,48,49,50]. Deletion of PU.1 in osteoclasts results in severe osteopetrosis with significantly fewer mature, multinucleated osteoclasts [51]. Several reports have demonstrated that MITF is a pivotal transcription factor involved in osteoclast differentiation [52,53,54]. Tal1/Scl1 (T-cell acute lyphocytic leukemia 1/stem cell leukemia 1), another basic helix-loop-helix transcription factor, is an important regulator of hematopoiesis [55] and osteoclastogenesis. PU.1 and MITF bind to the promoter region of DC-STAMP to activate its gene expression, whereas Tal1 works reciprocally with them as a transcriptional repressor [47]. Notably, suppressor Tal1 decreased when DC-STAMP was inhibited with microRNA-7b, suggesting that a negative feedback loop exists in this signaling pathway [56]. 

Tal1’s inhibitory effect on DC-STAMP can be attenuated by Strawberry Notch Homologue 2 (Sbno2). Sbno2-deficient mice exhibited increased bone mass due to impaired osteoclast multinucleation [57]. A recent study has additionally discovered that eomesodermin (EOMES) binds to the PU.1-MITF complex and facilitates its activity, and that EOMES knockdown decreases multinucleated osteoclast formation [58]. 

### 2.8. Several HDACs Are Involved in DC-STAMP Regulation

Histone deacetylase (HDAC) removes the acetyl group from the lysine residues of histones, which allows histone to wind chromatin up into a compact structure and suppresses the transcription of genes. Therefore, HDAC is usually considered a transcriptional repressor. With regard to osteoclast multinucleation, however, HDACs exhibit versatile functions [59]. 

HDAC2 (Class I) is essential for actin ring formation. The direct activation of Akt by HDAC2 abrogates FoxO1 (Forkhead box protein O1) [60], which negatively regulates osteoclastogenesis by reducing reactive oxygen species in mitochondria and inducing fusion of osteoclasts [61]. Although no direct interaction between HDAC2 and DC-STAMP has been shown, the expression of DC-STAMP is also downregulated by the inhibition of HDAC2 [60].

HDAC3 and HDAC7 have opposite effects on osteoclast multinucleation: HDAC3 is a positive regulator of multinucleation while HDAC7 is a negative regulator. HDAC7 suppresses the transcriptional activity of MITF and downregulates DC-STAMP [62].

HDAC5, HDAC10 and HDAC11 are upregulated during osteoclast differentiation under RANKL stimulation. Knocking down these HDACs increases the sizes of osteoclasts without affecting their numbers. HDAC5 and HDAC10 downregulate DC-STAMP through NFATc1, whereas HDAC11 can negatively affect DC-STAMP without changing the level of NFATc1 or c-Fos, indicating the existence of a direct or indirect relationship via other mechanisms between HDAC11 and DC-STAMP [63]. 

### 2.9. Proteins Interact with DC-STAMP

#### 2.9.1. Pin1 Directly Suppresses DC-STAMP

Islam et al. generated Pin1-deficient mice, which exhibited a phenotype with larger osteoclasts containing larger numbers of nuclei both in vitro and in vivo. They further showed that Pin1 directly binds and isomerizes DC-STAMP, attenuating its expression and altering its localization at the plasma membrane [64]. Cho et al. later found that *N*,*N*’-1,4-butanediylbis [3-(2-chlorophenyl) acrylamide] (BCPA) can attenuate the reduction of Pin1 expression during osteoclast differentiation, which suppresses DC-STAMP expression and osteoclast multinucleation [65].

#### 2.9.2. Luman, DC-STAMP and OS-9 Work as a Complex with Regard to Intracellular Localization and Activities

OS-9 and Luman are two binding partners of DC-STAMP whose union is important for their intracellular localization [66]. Under osteoclastogenic stimulation, the Luman–DC-STAMP–OS-9 complex might control the transport of Luman and DC-STAMP from the ER to the Golgi apparatus. After that, DC-STAMP redistributes to the plasma membrane, whereas Luman cleaves its N-terminus. This N-terminus of Luman moves to the nucleus and binds to the promoter region of DC-STAMP, enhancing its expression and thereby positively regulating osteoclast multinucleation [30].

### 2.10. Micro RNAs Directly Suppress the mRNA of DC-STAMP

Blocking DC-STAMP’s mRNA directly rather than blocking its protein by microRNA (miRNA) can further improve our understanding of the relevant factors. MiR-30a is downregulated under RANKL-stimulated osteoclastogenesis. It directly targets the mRNA (3′UTR) of DC-STAMP, decreasing its protein expression. Importantly, miR-30a further suppresses the protein expressions of c-Fos and NFATc1 without changing their mRNA levels by its inhibition of DC-STAMP [67]. MiR-7b also directly targets DC-STAMP [12,56]. This miRNA is also downregulated during osteoclastogenesis under RANKL and M-CSF stimulation, and its overexpression significantly abrogates the formation of multinucleated osteoclasts. The blockage of DC-STAMP is associated with low expression levels of other fusogens (OC-STAMP, ATP6v0d2, and CD47). c-Fos and NFATc1 are also known to be regulated by the downstream signaling of DC-STAMP. Collectively, these results confirm the significant role of the DC-STAMP-c-Fos-NFATc1 signaling pathway in the multinucleation of osteoclasts.

The regulation mechanisms and downstream molecules of DC-STAMP, as interpreted based on the findings summarized here, are illustrated in Figure 1. At present, the specific mechanism by which DC-STAMP regulates the osteoclast multinucleation process remains unknown. There is an urgent need to identify the ligand of DC-STAMP as this will enable us to clarify the overall mechanism.

### 2.11. DC-STAMP May Play a Role in Paget’s Disease of Bone

Accumulating studies have shed light on the association of DC-STAMP with Paget’s Disease of Bone (PDB), a disease with focal abnormalities of bone remodeling mainly caused by increased osteoclastic bone resorption. Albagha et al. identified DC-STAMP (*Tm7sf4*) as the possible effector of PDB through a genome-wide association study [68]. Beauregard et al. identified a rare variant (rs62620995) of DC-STAMP, although this turned out to have only a marginal association with PDB [69]. Laurier et al. later found that osteoclast-like cells from PDB patients carrying this variant showed higher expression levels of DC-STAMP and greater numbers of nuclei compared to both healthy controls and PDB patients without the variant, suggesting that these patients are more susceptible to PDB lesions [70]. Using cells from 158 PDB patients, Mullin et al. observed that a gain-of-function variant (rs2458413) was significantly associated with PDB [71]. Most recently, Sultana et al. identified a loss-of-function variant of DC-STAMP that co-segregated in a French-Canadian family with PDB. When this mutant DC-STAMP is transfected into HEK293 cells, the expression levels of AP1 and NF-κB are significantly reduced. Yet monocytes from PDB patients exhibit hyper-multinucleated phenotypes, and there is no significant association between this mutant DC-STAMP and PDB [72]. Although the direct association between DC-STAMP and PDB remains obscure, the association between DC-STAMP in human osteoclasts and bone remodeling is clear.

## 3. OC-STAMP

### 3.1. Protein Structure and Cellular Distribution of OC-STAMP Are Similar but not Identical to Those of DC-STAMP 

Paul R. Odgren’s group was the first to identify OC-STAMP using a microarray analysis of mouse primary bone marrow cells and RAW264.7 cells with/without RANKL stimulation. The gene encoding OC-STAMP is located on chromosome 2H3, and is strongly conserved in mammals including mice, rats and humans [16]. Odgren’s group later analyzed the protein structure in detail, revealing that OC-STAMP’s protein structure is similar but not identical to that of DC-STAMP. It penetrates the plasma membrane six times, and both the N- and C-termini are intracellular tails. There is a glycosylation site in the extracellular region that contributes to protein stability [14]. Initially, the expressions of OC-STAMP and DC-STAMP were confirmed in organs where cell fusion occurs, such as bone, muscle and skin [15]. Later, their universal expression was demonstrated in almost all human organs [73]. Notably, the expression of OC-STAMP is much lower in the ovary than in other organs, and later experiments have suggested that estrogen influences its expression [73]. 

### 3.2. Osteoclast Multinucleation Is Completely Inhibited in OC-STAMP-Deficient Mice, Which nevertheless Show Normal Growth and Skeletal Systems

Two studies have reported similar findings regarding the phenotypes of OC-STAMP knockout mice. In these mice, osteoclasts in bone tissue were mononuclear and multinucleation was completely blocked in vivo [14,15]. Even so, OC-STAMP-deficient mice showed normal skeleton, growth and bone metabolic markers, in marked contrast to the osteopetrotic phenotype of DC-STAMP knockout mice.

OC-STAMP-deficient cells isolated from bone marrow were able to differentiate into TRAP-positive osteoclasts under RANKL stimulation but could not fuse into multinucleated cells. The bone resorption ability of these mononuclear cells was about one-third that of wild-type cells [14]. The expression levels of other osteoclast markers (Cathepsin K, NFATc1, TRAP, DC-STAMP) were not significantly altered [14], indicating OC-STAMP’s specific role in osteoclast multinucleation rather than osteoclast differentiation.

### 3.3. Regulation of OC-STAMP Expression during Osteoclastogenesis

Little is known about the regulation of OC-STAMP. The fact that OC-STAMP expression in bone marrow monocytes increases only after stimulation with M-CSF and RANKL indicates that OC-STAMP is a RANKL-induced protein [73,74].

To date, however, specific knowledge about the regulation of OC-STAMP remains limited. As described above, Kim et al. found that its expression in the ovaries was lower than that in any other organ, and demonstrated that estrogen has a biphasic regulatory effect on OC-STAMP. Low concentrations of estrogen upregulate the expression of OC-STAMP whereas high concentrations downregulate it [73]. They proposed that OC-STAMP is involved in pathogenic bone resorption such as postmenopausal osteoporosis rather than normal bone metabolism. In addition, they administered NFAT activation inhibitor III to bone marrow monocytes and found that this did not influence the expression of OC-STAMP, suggesting that OC-STAMP is independent of NFATc1. They also demonstrated that inhibition of NF-kB, PKCβ (protein kinase C β), and Akt (also known as PKB, protein kinase B) completely blocked the expression of OC-STAMP independently from NFATc1 [73]. On the contrary, Miyamoto et al. administered FK506 (NFATc1 inhibitor) and found that this completely inhibited OC-STAMP [15]. Moreover, they identified two putative NFATc1 binding sites in the mouse OC-STAMP promoter region, and observed by means of a ChIP assay that NFATc1 binds to both sites [15]. The use of different inhibitors might be the cause of the conflict between these findings. In an earlier study, Huwang et al. reported that lysophosphatidic acid (LPA) can upregulate OC-STAMP through NFATc1, which also indicates a direct relationship between NFATc1 and OC-STAMP [75].

The downstream signals of OC-STAMP were also investigated through experiments using its specific antibody and shRNA [73]. OC-STAMP knockdown induced TRAF6 but decreased c-Jun, RANK, c-Fms and meltrin-α. Meltrin-α has also been reported to be involved in osteoclast multinucleation [76]. They proposed that OC-STAMP plays an important role in RANKL-RANK and M-CSF-c-Fms signaling, and that independent TNF-α (Tumor necrosis factor α)–c-Fms–TRAF6 signaling can restore osteoclastogenesis in the absence of OC-STAMP. A recent study has revealed that an OC-STAMP antibody specifically suppresses the mRNA and protein expression of CD9 along with osteoclast multinucleation [77]. Blocking DC-STAMP, on the other hand, does not inhibit CD9. Furthermore, antibodies to CD9 exert a synergic effect on the inhibition of multinucleation by antibodies to DC-STAMP, but not by those to OC-STAMP, suggesting that the OC-STAMP/CD9 axis promotes osteoclast multinucleation independent of DC-STAMP [77].

### 3.4. OC-STAMP and DC-STAMP Work Cooperatively

Knockout of either DC-STAMP or OC-STAMP inhibits osteoclast multinucleation. The fact that DC-STAMP expression is normal in OC-STAMP knockout cells and vice versa indicates that both are essential for osteoclast multinucleation [15]. There are three possible relationships between them [15]: (1) one is a ligand/receptor for the other; (2) they have distinct ligands, but each requires a positive reciprocal signal from the other for fusion to proceed; (3) they constitute a large receptor complex together. Further examinations are needed.

The reported molecules and signaling pathways related to OC-STAMP are schematically summarized in Figure 2. As in the case of DC-STAMP, that fact that we do not know the ligand of OC-STAMP prevents us from clarifying the whole picture of its downstream signals.

## 4. ATP6v0d2

### 4.1. ATP6v0d2 Is not Only a Proton Pump in Osteoclasts

Vacuolar ATPase (V-ATPase) is a giant molecule found in various parts of the endoplasmic reticulum system such as eukaryotic vacuoles, lysosomes, endosomes, and the Golgi apparatus, as well as in the cell membranes of osteoclasts, interneurons of the kidney assembly tract and some cancer cells. It acts as a proton pump coupled with ATP hydrolysis to transport protons in the intracellular space or across the cell membrane. In osteoclasts, V-ATPases in the plasma membrane can extracellularly release protons to the resorption lacunae, thus preserving the acidic extracellular environment needed for bone resorption [78,79,80]. V-ATPase consists of an extracellular domain V1 and a membrane-bound V0 domain. The V1 drives ATP hydrolysis to gain the energy for the rotation of the V0 domain and translocates protons across the membrane. Subunit d in the V0 domain has two isoforms, d1 and d2. In mature osteoclasts, the expression of ATP6v0d1 is up to 10 times higher than that of ATP6v0d2 [81,82].

In addition to its role as a proton translocator, ATP6v0d2 plays an essential role in intracellular autophagosome-lysosome fusion in macrophages [83] and in osteoclast fusion [84]. The expression of ATP6v0d2 increases during osteoclast differentiation, reaching its highest level in mature osteoclasts. The phenotype of ATP6v0d2 knockout mice is very similar to that of DC-STAMP knockout mice, showing increased bone mass of the tibial bone. The total number of TRAP-positive cells is not altered, but that of multinucleated TRAP-positive cells is markedly reduced in vivo. Bone marrow monocytes from ATP6v0d2-deficient mice failed to multinucleate, yet preserved the same levels of osteoclast markers such as Oscar, Ctsk, and Acp5 [84]. However, ovariectomy (OVX) in ATP6v0d2-deficient mice suppressed osteoclast multinucleation, although the quantity of osteoclast precursors increased. ATP6vd2 deficiency could not attenuate osteoporosis induced by OVX [85]. The researchers who observed this suggested that, in pathogenic osteoporosis, the suppression of osteoclast multinucleation might be masked by an increase in osteoclast precursors.

### 4.2. How Is ATP6v0d2 Regulated and Involved in Osteoclast Multinucleation?

A binding site of NFATc1 has been identified in the promotor region of ATP6v0d2. The amplification of NFATc1 expression by RANKL-RANK-, LPS (Lipopolysaccharide)- or TNF-α-derived signals directly promotes the expression of ATP6v0d2, and contributes to osteoclast multinucleation [31]. Feng et al. further found that extrinsic MITF and NFATc1 migrate into the nucleus and co-operatively bind to the promotor region of ATP6v0d2 with nuclear myocyte enhancer factor 2 (MEF2) [86].

The mechanism by which ATP6v0d2 regulates osteoclast multinucleation is unknown. Kim et al. identified adhesion-regulating molecule 1 protein (Adrm1) as a potential functional partner using yeast two-hybrid screening [87]. Intriguingly, this adhesion-regulating protein did not affect the cell adhesion of OCs but was required for cell migration and OC multinucleation. Knockdown of Adrm1 significantly reduced the fusion efficiency of preosteoclasts without affecting other known RANKL-induced pathways including NF-kB and MAPK [87].

Li et al. have shown that the functions of ATP6v0d2 in osteoclasts are different from those of other osteoclastogenic molecules [88]. N6-adenosine-methyltransferase 70 kDa subunit (METTL3), an enzyme involved in the post-transcriptional methylation of internal adenosine residues in eukaryotic mRNAs, has a different influence on ATP6v0d2 from that of other osteoclast-specific genes [88]. Depletion of Mettl3 inhibited the expression of Nfatc1, c-fos, Ctsk, Acp5 and Dc-stamp, but significantly upregulated Atp6v0d2 by stabilizing its mRNA via YTHDF2 (YTH-Domain Family Member 2, an m6A-binding protein that recognizes and destabilizes m6A-modified mRNA [89]) participation [88]. Osteoclasts differentiated from Mettl3-deficient cells were smaller in number but were larger in size and had more nuclei. Of note, these Mettl3-deficient cells exhibited lower bone resorption activity in spite of their increased size, indicating that ATP6V0d2-mediated multinucleation is independent of osteoclastogenesis.

### 4.3. P-ATPase α1 and NHE10 also Regulate Osteoclast Multinucleation

Two other ion pumps have also been reported to regulate osteoclast multinucleation. RANKL-induced Na+/K+-ATPase (P type ATPase) α1 [90] and Na+/H+ exchangers (NHE10) [91] positively regulate osteoclast fusion. Although these, unlike ATP6v0d2, seem to depend on the RANKL-DC-STAMP axis, it is reasonable to consider that the ion pumps, or perhaps their resulting pH environments, not only are associated with bone resorption but also affect osteoclast multinucleation.

## 5. CD47-MFR

Macrophage fusion receptor (MFR, also known as SHPS-1 or SIRPα), a transmembrane immunoglobulin, was identified as a fusion protein in macrophages by Saginario et al. in the last century [92,93]. The transmembrane glycoprotein CD47 has been identified as its ligand, and their interaction is essential for the onset of osteoclast fusion [94]. Blocking the CD47/MFR pathway strongly reduced the formation of TRAP+ multinucleated osteoclasts in vitro, and CD47-deficient mice showed decreased numbers of osteoclasts and osteoblasts, which resulted in an osteopenic bone phenotype [95,96]. The ligation of CD47 induces tyrosine phosphorylation of MFR’s cytoplasmic tyrosine residue (ITIM), which recruits and activates the SH2-containing protein tyrosine phosphatase-1 (SHP-1) and subsequently dephosphorylates non-muscle cell myosin IIA (Myf9) [97]. Myosins belong to the superfamily of actin-based motor proteins. In osteoclasts, myosin II appears to play a contractile function, regulating the movement of clustered podosomes [98]. Myosin IIA, which also generates contractile forces in the osteoclast cytoskeleton, must be downregulated at the onset of cell fusion [99].

CD47 was found to be involved primarily in the cell fusion of small osteoclasts in the early stages of osteoclastogenesis and mononucleated pre-osteoclasts [100,101], but it seems not to be the only ligand for MFR. A recent study has revealed that blocking MFR with a neutralizing antibody completely inhibits osteoclastogenesis of bone marrow cells even under RANKL and M-CSF stimulation [102], which did not occur when CD47-MFR signaling was blocked. Another ligand, CD36, also expresses on the surfaces of preosteoclasts and participates in MFR-mediated osteoclastogenesis. CD36 and CD47 double-knockout osteoclasts showed impaired resorptive activity in vitro. Surprisingly, however, double-knockout mice did not have a significantly abnormal bone phenotype [102], whereas CD36-deficient mice and MFR-deficient mice both showed an osteopetrotic phenotype [102,103]. Therefore, other unknown factors seem to be involved in MFR-mediated osteoclast multinucleation. Podolnikova et al. recently reported that integrin macrophage antigen 1 (Mac-1) expressed on macrophages is a counter-receptor for MFR, and that their interaction is involved in macrophage fusion [104].

## 6. Cadherins

The cell adhesion molecules known as cadherins are categorized into classical cadherins (E-cadherin, N-cadherin, etc.), desmosomal cadherins and protocadherins. Two cadherins from different categories have been reported to be involved in osteoclast multinucleation.

Mbalaviele et al. were the first to demonstrate the presence of E-cadherin in differentiating osteoclasts; they also reported that blocking E-cadherin function reduced TRAP-positive multinucleated cell numbers [105]. Fiorino et al. further revealed that E-cadherin is expressed at the early stages of osteoclastogenesis and is mainly distributed at cell contact points and the polarized regions of cell membranes. They suggested that E-cadherin is involved in the formation of dynamic membrane protrusions, and that blocking E-cadherin impairs the migratory behaviors of fusing pre-osteoclasts, which results in attenuated cell fusion, rather than impairing the fusion process itself [106]. Interestingly, when RAW264.7 cells were indirectly co-cultured with carcinoma cells, Quan et al. found that E-cadherin decreased in carcinoma cells, yet simultaneously increased in differentiating pre-osteoclasts, and that larger osteoclasts were formed [107]. It does not seem likely that E-cadherin is directly involved in the osteoclast multinucleation process; nevertheless, its presence seems to be essential for the process.

Protocadherin 7 is the second cadherin that appears to be required for osteoclast multinucleation [108]. Its gene Pcdh7 was identified through a global epigenomic analysis that searched for genes with a histone modification from H3K4me3(+)H3K27me3(+) to H3K4me3(+)H3K27me3(−) during osteoclast differentiation. This histone modification also occurs in Nfatc1 in osteoclastogenesis [108,109]. Pcdh7 is regulated by NFATc1, and the expression levels of Mitf, Dcstamp, Ocstamp and Atp6v0d2 were significantly decreased by Pcdh7 knockdown, inhibiting the formation of TRAP+ multinucleated cells with more than nine nuclei.

## 7. Syncytins

Syncytin is a human endogenous retroviral gene that was integrated into the human genome millions of years ago. The encoded proteins (syncytin-1 and syncytin-2 in humans, and syncytin-A and syncytin-B in mice) mediate membrane fusion; they were originally involved in the binding of a virus to the host membrane and in the fusion of the membranes between virus and host. In the human body, this protein still mediates membrane fusion, particularly in placenta formation [110,111]. System alanine-serine-cysteine transporter 2 (ASCT2) has been identified as its receptor. Ligation of syncytin-1 to ASCT2 changes the conformation of the transmembrane subunit of syncytin-1 so that it extends a fusion peptide into the opponent cell membrane, inducing subsequent merging of the membranes [112].

Soe et al. first noticed the involvement of syncytin-1 in human osteoclast fusion in vitro [113]. Taking advantage of osteoclast precursors isolated from the blood of healthy female volunteers, they showed that M-CSF induced both syncytin-1 and ASCT2, while the addition of RANKL decreased syncytin-1 but not ASCT2. Syncytin-1 is involved in the formation of multinuclear osteoclasts with more than four nuclei but not osteoclasts with fewer than three, suggesting its involvement in the late phase of fusion. The proteins’ distributions are also intriguing. Syncytin-1 expresses in the plasma membranes of fusing osteoclasts, including cellular structures such as the podosome, filopodia and lamellipodia. The distribution of syncytin-1 surprisingly overlaps with that of actin, further indicating the former’s involvement in cytoskeleton formation and membrane activities. Soe et al. also observed an alteration of syncytin-1 distribution during the fusion process. They identified four types of cell-cell contact prior to fusion, based on the morphological appearance of contact: broad contact surface, phagocytic cup-like, tube contact and tip of filopodia [114]. They found that the members of each pair of approaching fusion partners exhibit heterogeneity in the localization of syncytin-1 in the plasma membrane. In phagocytic cup-like cell contact, as an example, one of the approaching osteoclasts distributed syncytin-1 to the membrane facing the other osteoclast such that it resembled a phagocytic cup, whereas the other osteoclast expressed a very low level of syncytin-1 that was allocated far from the contact area [100,113]. The same group later used time-lapse imaging to show that syncytin-1 mediates the fusion of multinucleated osteoclasts rather than that of two mononucleated pre-osteoclasts or that of mononucleated pre-osteoclasts with multinucleated osteoclasts. Furthermore, syncytin-1 mainly mediates fusion through phagocytic-cup like cell-cell contact [101].

Recently, another specific molecular mechanism of syncytin-1-mediated osteoclast fusion was reported [115]. The beginning of osteoclastic cell fusion was synchronized by administering lysophosphatidylcholine to pause the RANKL-stimulated ready-to-fuse osteoclast precursors. The blocking of RANKL-RANK signaling by antibodies while also removing the fusion inhibitor did not inhibit initiation of the fusion process, though the addition of antibodies against DC-STAMP and syncytin-1 did abrogate osteoclast fusion. These results suggest that RANKL-RANK signaling is essential to the initiation of osteoclastogenesis and subsequent cell fusion, but is not required for the fusion process itself, whereas DC-STAMP and syncytin-1 are the key regulators during the fusion process [115]. The same researchers further revealed that, during the fusion process, non-apoptotic phosphatidylserine (PS) externalization occurs in the lipid bilayer plasma membrane depending on the activity of DC-STAMP. The PS-binding protein Annexin A5 and the S100-family calcium-binding protein S100A4, which binds to Annexin A5, are also required for membrane fusion. The research group proposed a membrane fusion model in which receptor-triggered Syncytin-1 extends its fusion peptide and assembles the Annexin A5-S100A4 complex at the externalized PS, constructing a lowest-energy hairpin conformation to anchor and merge the membranes. Conflicting results, however, were reported in Irie et al.’s research showing that phosphatidylethanolamine (PE) is observed during osteoclast fusion and is essential to the process, but the externalization of PS to the outer leaflet of the plasma membrane bilayer is not [116]. Further research is needed to resolve the apparent discrepancy between these results.

The in-vivo behavior of syncytin in osteogenesis was observed by Coudert et al. in 2019 [117]. The phenotypes of syncytin-B knockout mice were investigated because of the embryonic lethality of syncytin-A knockout mice. Of note, syncytin-deficient mice did not display any malformation of osteoclasts or altered bone phenotype, although syncytin deficiency inhibits multinucleation ex vivo. Conditional knockout of syncytin-A, or double conditional knockout of syncytin-A and syncytin-B, is required to further clarify the underlying mechanism.

## 8. ITAM-Bearing Immunoreceptors

DAP12 and FcRγ are two ITAM-bearing immunoreceptors that co-stimulate osteoclastogenesis along with RANKL-RANK signaling [118]. These immunoreceptors do not possess ligation domains, so they pair with other immunoreceptors such as OSCAR [119], TREM2 [120], and Siglec-15 [121] to receive signals from extracellular ligands. These immunoreceptors also play an important role in osteoclast multinucleation through ITAM signaling.

### 8.1. OSCAR-FcRγ

OSCAR in osteoclasts was first described in 2002 by Kim et al., who identified it as an immunoreceptor that is necessary for osteoclast maturation, including multinucleation [122]. OSCAR only expresses on osteoclasts in mice, but human OSCAR is also expressed on monocytes and macrophages, neutrophils and myeloid dendritic cells [123]. OSCAR is a receptor for extracellular collagens. It binds to a specific triple-helical motif in collagen, co-stimulates with FcRγ, and induces intracellular signaling [119]. OSCAR-FcRγ signaling seems to be pivotal in allowing pre-osteoclasts in the collagenous bone marrow space to become fusion-competent. In mature osteoclasts on bone surfaces, on the other hand, the expression of OSCAR decreases [124]. Furthermore, the collagenous environment in bone marrow, which may strengthen signaling through OSCAR, is reported to be favorable for pre-osteoclast fusion [124].

### 8.2. DAP12

ITAM signaling of DAP12 has been shown to program macrophages into a fusion-competent state [125] and to be critical to the regulation of OC multinucleation and migration [119,120]. DAP12 is a type I transmembrane protein expressed as a disulfide-bonded homodimer on natural killer cells, myeloid cells and a subset of T cells. DAP12 has a very small extracellular region, a transmembrane domain containing an aspartic acid residue that is critical for association with its partner subunits, and an intracellular domain with a single ITAM [126]. DAP12-deficient bone marrow-derived monocytes/macrophages fail to differentiate into osteoclasts, but if these precursors are co-cultured with osteoblasts, they can differentiate into TRAP-positive osteoclasts, although multinucleation is totally blocked. The bone phenotype of DAP12 knockout mice shows only mild osteopetrosis [127,128]. Taken together, these findings suggest that DAP12 is essential for osteoclast multinucleation, but its role in regulating osteoclast differentiation can be filled by other molecules in its absence. 

### 8.3. Siglec-15

Siglecs (sialic acid-binding immunoglobulin-type lectins) are plasma membrane-bound proteins. Takahashi et al. identified siglec-15 in immune cells, where it regulates cell functions through glycan recognition and interacts with DAP12 [129]. Hiruma et al. later reported its overexpression in TRAP-positive multinucleated giant cells of giant cell tumor of bone (GCT), and demonstrated that siglec-15 is also a key player in osteoclastogenesis [130]. 

Siglec-15 is upregulated by RANKL in mouse primary bone marrow macrophages [131]. Migrating polykaryons, which are characterized by multiple nuclei and lamellipodia, express an especially high level of Siglec-15 on the plasma membrane, whereas mononuclear TRAP-positive cells exhibit only a low expression level [131]. 

Siglec-15-deficient mice showed a mild osteopetrosis phenotype [131,132]. Kameda et al. demonstrated that the numbers of multinucleated osteoclasts were significantly lower in the secondary spongiosa in Siglec-15-deficient mice. In the primary spongiosa, they suggested, collagen type II is abundant, and OSCAR-FcRγ signaling may compensate for siglec-15 deficiency [131]. Hiruma et al., in contrast, found no reduction in the numbers of osteoclasts in vivo, but both groups observed impaired osteoclast multinucleation in Siglec-15-deficient bone marrow cells in vitro [132].

### 8.4. Siglec-15/DAP Association

The phenotype of siglec-15-deficient mice is similar to that of DAP12-deficient mice. DAP12 is an activating adaptor protein with a small extracellular domain that cannot bind to ligands. Therefore, DAP12 pairs with DAP12-associated receptors (DAR) for ligation and subsequent intracellular signaling [129]. Siglec-15 is one of these DARs, and signaling through Siglec-15-DAP12 is demonstrated to be important for osteoclast multinucleation and cytoskeleton formation [121,129]. 

The signaling pathway involving Siglec-15/DAP12 in osteoclasts has been partially explored. CD44 on the cell surfaces of pre-osteoclasts has been identified as the relevant ligand [133]. After ligation, spleen tyrosine kinase (SYK) is recruited through ITAM signaling of DAP12 [134], which also enhances the phosphorylations of ERK and PI3K/Akt under RANK-TRAF6 signaling [131]. Siglec-15/DAP12 cross-talks with RANK-TRAF6 through ERK and PI3K/Akt, but the expressions of NFATc1 and NF-κB are not altered [121,131]. In addition, TNF-α-induced osteoclastogenesis was also blocked when Siglec-15 was deficient, though the expression levels of ERK and PI3K/Akt were unchanged [135]. The expression of DC-STAMP was likewise not altered in Siglec15-deficient osteoclasts [135]. PI3K and Akt are essential for the formation of the actin ring [136,137]; indeed, Siglec-15-deficient osteoclasts failed to form actin rings. Actin dynamism is important for the formation of the cytoskeleton, including filopodias and podosomes, which are involved in osteoclast fusion [138,139,140]. The regulation of cytoskeleton development might influence subsequent multinucleation.

Siglec-15 is not the only associate receptor of DAP12. In Siglec-15-deficient pre-osteoclasts under RANKL stimulation, osteoclast markers such as TRAP, Cathepsin K, integrin β3 and calcitonin receptor are inhibited, though the expression of NFATc1 is not altered [131]. Given that DAP12 is essential for NFATc1 induction through calcium signaling [128], other associated receptors such as TREM2 may work cooperatively with Siglec-15 to enable signal transduction. 

### 8.5. Treatment of Osteoporosis Targeting Siglec-15

Stuible et al. administered a monoclonal antibody against Siglec-15 to young mice, which developed increased bone mineral density as a result [141]. In vitro, the antibody totally inhibited the multinucleation of human peripheral blood-derived pre-osteoclasts under seven days of RANKL stimulation. A prolonged period of RANKL stimulation finally induced multinucleation, but the secretion of TRAP and resorption ability were greatly decreased by antibody administration [141].

Takahata’s group performed ovariectomies on Siglec15-deficient mice, resulting in attenuated osteoporosis due to impaired osteoclast function [135]. The same group later compared the effects of anti-Siglec-15 antibody and alendronate [142]. Anti-Siglec-15 therapy and alendronate both increased bone mass to a comparable degree, but anti-Siglec-15 antibody did not affect the skeletal growth of young mice as alendronate did [142]. A more recent study from the same group further investigated the same effects in glucocorticoid-induced osteoporosis without affecting chondrocytes and osteoblasts [143]. The evidence collectively indicates that Siglec-15-targeting therapy may specifically regulate osteoclast function without diminishing bone formation. 

## 9. NETRIN-1, Flrt2 and Unc5b axis

Osteoblast- and synovial cell-derived Netrin-1 ligates to uncoordinated-5b receptor (Unc5b), enhancing the expression of ITIM-harboring receptors such as paired immunoglobulin-like receptor (PIR-B) and MFR/SIRPα, which can activate the tyrosine phosphatase known as Src homology region 2 domain-containing phosphatase-1 (SHP-1) [144,145]. SHP-1 is reported to be a negative regulator of multinucleated osteoclast formation [146]. SHP-1 is a phosphatase of the VAV family, in which VAV3 is a key downstream component of integrin β_3_ signaling and an upstream activator of Rho GTPases such as Rac1 [6,147]. Rho GTPases regulate actin dynamism and, subsequently, the cytoskeleton, including the formation of podosomes and filopodias. Actin flow in the zipper-like structure formed by the podosome belts of two contacting osteoclasts is involved in osteoclast fusion [140]. Filopodias are long, thin, actin-rich cellular protrusions that initiate osteoclast fusion [138]. Collectively, Netrin-1’s ligation to Unc5b negatively regulates osteoclast fusion through SHP-1-VAV3-Rac1 signaling [148]. 

Osteoclast-derived fibronectin leucine-rich transmembrane protein 2 (Flrt2) is an antagonist of Netrin-1 in osteoclasts. Flrt2 knockout mice exhibit few multinuclear osteoclasts both in vivo and in vitro, and blocking Unc5b restores osteoclast fusion. In addition, administration of Netrin-1 to Flrt2-/-osteoclast precursors does not induce any additional inhibitive effects on multinucleation, suggesting that Unc5b receptors are fully occupied by endogenous netrin-1 without the existence of Flrt2. Taking these findings together, we can conclude that Flrt2 and Netrin-1 work cooperatively to fine-tune osteoclast multinucleation [149].

## 10. Transglutaminases

Transglutaminases (TGs) are enzymes that mediate covalent cross-linking between glutamine residues and various primary amines in a calcium-dependent manner through the post-translational modification of proteins [150]. There are eight identified active TGs: TG1-7, as well as factor XIII A [150]. Tanaka et al. have demonstrated that the activity of TGs is necessary for macrophage fusion [151]. Their factor XIII A-deficient mice were resistant to bone destruction due to collagen-induced arthritis and had fewer multinucleated osteoclasts both in vivo and in vitro. Furthermore, RANKL expression was downregulated in their fXIIIA-/- mice, indicating a positive role for factor XIII A in osteoclastogenesis [152]. Another group, however, found that factor XIII A and TG2 double-knockout mice exhibited an osteopenic phenotype with increased osteoclastogenesis, suggesting negative roles for these two molecules [153]. Kim et al. also showed that TG2 knockdown enhanced osteoclast multinucleation by upregulating Blimp1 and thereby suppressing Bcl6, which is a transcriptional repressor of NFATc1 and DC-STAMP [154]. Interestingly, a pan-inhibitor of TGs blocked osteoclastogenesis in factor IIIA and TG2 double-knockout pre-osteoclasts, which further confirmed TG1 as another regulator in osteoclastogenesis [153]. Their recent work demonstrated that TG1, TG2 and factor XIII A are expressed during osteoclast differentiation, and that their activities, especially that of factor XIII A, reached their highest levels in the early fusion phase and decreased as osteoclasts matured. These TGs co-localize to podosomes. Blocking TGs would impair the actin dynamics of osteoclasts, thereby inhibiting the migration and fusion of pre-osteoclasts [155,156].

## 11. Purinergic Receptors

Purine receptors are a group of cell surface receptors whose ligands are nucleotides such as adenosine and ATP. They are classified into the P1 and P2 receptor families; P1 is an adenosine receptor and P2 is an ATP receptor. P2X receptors are ion channel receptors that, when ligated by extracellular ATP, can open large pores in the plasma membrane for the transportation of giant molecules such as cytokines and nucleotides [157]. P2X7 and P2X5 have been thoroughly studied with regard to osteoclastogenesis.

P2X7 in osteoclasts seems to function differently in vitro and in vivo. Initially, P2X7 was reported to be necessary for macrophage fusion and the formation of multinucleated FGBCs in vitro [158]. Soon after, however, the regulatory role of P2X7 in multinucleation of pre-osteoclasts was indicated in both mice [159] and humans [160,161]. The blockage of P2X7 signaling by antibodies or antagonists inhibits osteoclast multinucleation in vitro, but the details of the mechanism of this effect remain largely unknown. Hiken et al. treated RAW264.7 cells with prolonged exposure to ATP and found that P2X7 disappeared from the cell surfaces and became resistant to ATP stimulation. These ATP-resistant cells could not fuse, even under RANKL stimulation [162]. Their finding further demonstrated the necessity of ATP-P2X7 signaling for osteoclast multinucleation but still left the precise mechanism unclear. Hwang et al. demonstrated that lysophosphatidic acid (LPA) can stimulate the expressions of P2X7 and OC-STAMP via NFATc1 and thus can control the multinucleation of murine primary osteoclasts [75]. Pellegatti et al., however, suggested a surprising explanation for this: they revealed that fusion-competent pre-osteoclasts release endogenous ATP into the extracellular matrix via the P2X7 receptor, and that the degradation of extracellular ATP into adenosine, along with its P1 receptors (A1/A2A), is dispensable for osteoclast multinucleation [163]. Knowles also confirmed that osteoclast fusion in a hypoxic or inflammatory environment requires sufficient quantities of extracellular adenosine and its P1 receptor [164]. P2X7 receptor has also been reported to regulate the secretion of TG2, which is another regulator of osteoclast multinucleation [165,166]. The question of whether P2X7 receptor itself is involved in the fusion process as opposed to indirectly influencing other signaling pathways remains open.

The bone phenotypes of P2X7-deficient mice range from osteoporosis to slightly increased bone mass, but all have normal multinucleated osteoclasts in vivo, as shown in various reports and strains of mice [159,167,168]. Moreover, these reports consistently state that primary bone marrow cells from P2X7-/- mice can normally form multinucleated osteoclasts, strongly denying the necessity of P2X7 for osteoclast multinucleation. Human genotyping studies have also been conducted. The single nucleotide polymorphism (SNP) causing loss-of-function of P2X7 accelerates bone loss [169] and increases fracture risk [170] in post-menopausal women, indicating a protective role of P2X7 in bone metabolism. As an explanation for this, P2X7 was reported to be involved in the apoptosis of cells including osteoclasts [170] as well as being expressed in osteocytes and osteoblasts and positively regulating their functions [171,172,173]. The balance of bone remodeling achieved by these cells can be finely tuned by P2X7 in vivo.

A series of studies has recently reported that another P2X receptor, P2X5, is required for osteoclast multinucleation [174,175,176]. Kim et al. found that P2X5 is highly expressed during the fusion phase of multinucleated osteoclasts and multinucleated giant cells, and that blocking P2X5 signaling with an antagonist or ATP diphosphatase apyrase inhibits multinucleation and resorptive ability but not osteoclast differentiation. In addition, bone marrow cells from P2X5-deficient mice cannot fuse under RANKL stimulation. They further revealed that this blockage is independent of NFATc1, c-Src or Atp6v0d2, while the expression levels of OSCAR, Ctsk, carbonic anhydrase 2 and integrin b3 are downregulated [176]. In vivo, they used an LPS-induced calvarial bone destruction model [176] and a ligature-induced periodontitis model [175], and in both cases confirmed that P2X5 deficiency has a protective effect against bone loss. They identified decreased cytokine expressions in P2X5-deficient tissues, suggesting an association of P2X5 with inflammatory osteoclast activation [175]. Their latest work has further identified methylosome protein 50 (MEP50), a cofactor of protein arginine methyltransferase 5 (PRMT5) that associates with P2X5’s intracellular C-tail for osteoclast multinucleation. Their future studies deserve our continued attention.

## 12. TRAIL

Human TNF-related apoptosis-inducing ligand (TRAIL), another member of the TNF superfamily of cytokines, not only induces apoptosis in several tumors but is also reported to trigger osteoclastogenesis through TRAF6 activity, independently from RANKL-RANK signaling [177,178]. TRAIL alone at a relatively high concentration (100ng/mL) can induce multinucleated osteoclast formation from human monocytes and the murine RAW264.7 cell-line [178]. 

Medaka fish in a space station showed enhanced osteoclastogenesis [179]. Two unique in-vitro experiments based on this finding revealed that microgravity and radiation affect the expression of TRAIL, which enhances osteoclast formation. The researchers used a rotating culture system to create microgravity on Earth. RAW264.7 cells under microgravity conditions expressed high expression levels of TRAIL, and subsequent high expression levels of TRAF6, OC-STAMP and DC-STAMP, in the absence of RANKL. Moreover, under RANKL stimulation, TRAIL-neutralizing antibodies attenuated the formation of multinucleated osteoclasts, suggesting a regulative role of TRAIL on osteoclast multinucleation. These experiments did not assess other osteoclastic markers such as c-Fos and NFATc1, which are activated by TRAF6 and positioned upstream of the STAMPs, leaving the exact relationship between TRAIL and osteoclast multinucleation ambiguous [180,181]. In addition, in spite of their intriguing findings, several lines of evidence suggest that TRAIL preferentially inhibits the RANKL-RANK pathway of osteoclast precursors under RANKL stimulation [182], which conflicts with the idea that the blockage of TRAIL can attenuate osteoclastogenesis.

## 13. Recent Pre-Clinical Studies

In the last ten years, in addition to identifying ever more molecules that are involved in osteoclast multinucleation, several research groups have investigated various chemical compounds related to osteoclast multinucleation (Table 1).

Pregnenolone, a prohormone of most of the steroid hormones including estrogen, glucocorticoids and progestogens, inhibits multinucleation of osteoclasts and their bone resorptive ability in vitro, and exerts protective effects against osteoclasts in LPS-induced inflammatory bone destruction and ovariectomy-induced bone loss in vivo. Pregnenolone exhibits antioxidant properties in osteoclasts and inhibits ERK, MAPK and NF-kB activation, thereby impairing c-Fos and NFATc1, which results in downregulated expressions of fusogens such as DC-STAMP and ATP6v0d2 [183]. 

Raw *Pisidium coreanum* is used in traditional Chinese medicine. Choi et al. have recently demonstrated its ability to reduce osteoclast size and resorptive ability both in vitro and in vivo and to increase bone mass in an OVX mouse model. Gene expressions of NFATc1, p65, and integrin αvβ3 were significantly inhibited, and those of fusion-related genes such as *DC-STAMP*, *OC-STAMP*, *Atp6v0d2*, *FAK*, *CD44*, and *MFR* were also significantly reduced by the treatment [184].

Sulforaphane is a kind of isothiocyanate found in cruciferous vegetables such as broccoli, Brussels sprouts, and cabbages. Multiple studies have indicated its negative regulation of osteoclast multinucleation [185,186,187]. Investigations into the mechanism for this effect have shown that, under sulforaphane treatment, the immunoreceptor OSCAR is upregulated, which further induces the phosphorylation of STAT1, which in turn inhibits the expressions of *DC-STAMP* and *OC-STAMP* [187].

Extracts from barley seedlings, a natural food, have also been shown to prevent osteoclast fusion. When differentiating mice bone marrow monocytes are treated with barley seedling extracts on day 3 to day 4, the expression level of DC-STAMP and cell fusion are markedly repressed [188].

Aconine, a traditional Chinese medicine, has been shown to inhibit osteoclastogenesis in vitro and to suppress β3-integrin and DC-STAMP through the downregulation of NFATc1 and NF-kB, leading to inhibited osteoclast fusion [189]. 

Melphalan, used as a cytotoxic chemotherapy drug, increases the expressions of the fusion factors DC-STAMP and OC-STAMP by upregulating MITF rather than NFATc1, thereby promoting osteoclast multinucleation [190]. 

High concentrations of extracellular inorganic phosphate (Pi), which are often caused by chronic kidney disease, strongly inhibited the expression of c-Fos. C-Fos is a component of transcription factor AP-1, which directly binds to the promoter region of DC-STAMP along with NFATc1. High Pi therefore inhibits osteoclast fusion by reducing DC-STAMP expression [191].

Tsukamoto et al. used elcatonin (modified eel calcitonin) to treat murine osteoporosis induced by a tail suspension model. Elcatonin treatment significantly improved BMD (Bone Mineral Density) and BV/TV (Bone Volume/Tissue Volume), and, notably, decreased the numbers of multinuclear TRAP-positive osteoclasts while increasing those of mononuclear TRAP cells, suggesting its inhibitive effect on multinucleation. The same authors found that elcatonin treatment also suppressed the mRNA expression levels of NFATc1, Ctsk, and ATP6V0D2 [192].

Dou et al. found that cyanidin exerts a biphasic effect on osteoclast fusion: low doses can promote fusion while high doses inhibit it. They further showed that cyanidin dose-dependently regulates almost all the known osteoclast fusogens, including CD9, CD47, ATP6v0d2, DC-STAMP, OC-STAMP and OSCAR [193].

Kuriya et al. recently found that epigallocatechin gallate, a tea extract, specifically suppresses the expression of DC-STAMP in RAW264.7 cells and inhibits osteoclast multinucleation [194]. The TRAP activity of cells treated with epigallocatechin gallate is upregulated, further suggesting a specific function controlling osteoclast multinucleation without affecting the differentiation of DC-STAMP.

Kanzaki et al. have reported that phosphoglycerol dihydroceramide (PGDHC), a ceramide produced by the peridontal pathogen *Porphyromonas gingivalis*, is the virulence factor promoting the formation of larger osteoclasts. PGDHC directly binds to non-muscle myosin IIA (Myh9), elevating the expression of DC-STAMP by activating Rac1 GTPase [195].

Transient muscle paralysis induced by the injection of botulinum toxin A (BTxA) causes local bone resorption. In connection with this process, Ausk et al. have observed rapid inflammatory cell infiltration and upregulation of inflammatory cytokines such as TNF-α and IL-1, followed by high expression levels of DC-STAMP and OC-STAMP, suggesting the involvement of the STAMPs in inflammation [196]. That study did not investigate the exact connections between these factors, however.

Vitamin E (α-tocopherol), despite its antioxidant properties, has been shown to have a pro-bone resorption effect by promoting osteoclast multinucleation through DC-STAMP [197]. Mice deficient in α-tocopherol transfer protein (Ttpa-/-), which have reduced serum α-tocopherol concentrations, exhibit an osteopetrotic phenotype as a result of their smaller osteoclasts and lower bone resorption rates. In accordance with these in-vivo findings, administering α-tocopherol to wild-type mouse bone marrow cells enhanced the phosphorylation of p38 in the MAPK cascade, which further activated the expression of Mitf, a transcription factor of DC-STAMP, and thereby enhancing osteoclast multinucleation. In addition, wild-type mice fed with α-tocopherol (600 mg/kg) for eight weeks showed a 20% decrease in bone mass. Given that α-tocopherol supplements are consumed by many people worldwide, this finding is of great importance. Yet, in contrast to these results, later reports have shown that excessive intake of vitamin E did not cause bone loss in mice (200–1000 mg/kg) [198] or rats (600 mg/kg) [199]. In humans, the effect of vitamin E on bone health is likewise controversial. Zhang et al. identified a negative association between serum α-tocopherol concentration and bone mineral density in an elderly US population [200], and Ilesanmi-Oyelere et al. reported that a diet rich in vitamin E and fats was associated with lower bone mineral density in postmenopausal women [201]. On the other hand, Odai et al.’s cross-sectional study showed that dietary intake of vitamin E was associated with higher bone mineral density in premenopausal women but had no effect in postmenopausal women [202]. Yang et al. found no effect of vitamin E on bone turnover markers or bone mineral density in peri- or postmenopausal women in the UK [203]. The antioxidant properties of vitamin E and their beneficial effects on other bone cells such as osteoblasts might be the confounding factors leading to these conflicting results [204].

**Table 1 ijms-21-05685-t001:** Recent studies of various compounds targeting osteoclast multinucleation.

Author and Year of Publication	Compound Used	In-Vitro Cell Types	In-Vivo Model	Affected Molecules
Sun 2020 [183]	Pregnenolone	Mouse bone marrow macrophages	LPS-induced calvarial osteolysis model of mice	MAPK, NF-κB, c-Fos, NFATc1, Cathepsin K, Trap, DC-STAMP, ATP6v0d2
Choi 2019 [184]	*Pisidium coreanum*	Mouse bone marrow macrophages	OVX mice treated with *Pisidium coreanum*	NFATc1, p65, integrin αvβ3, DC-STAMP, OC-STAMP, Atp6v0d2, FAK, CD44, and MFR
Takagi 2017 [187]	Sulforaphane	Mouse bone marrow cells RAW264.7	-	OSCAR, NFATc1, Trap, Cathepsin K, DC-STAMP, OC-STAMP
Choi 2017 [188]	Barley seedling extracts	Mouse bone marrow cells	-	IκB, c-Fos, NFATc1, DC-STAMP
Zeng 2016 [189]	Aconine	RAW264.7	-	NF-κB, NFATc1, DC-STAMP
Chai 2017 [190]	Melphalan (increased multinucleation)	Mouse bone marrow cells RAW264.7	Mice treated with melphalan	Mitf, DC-STAMP, OC-STAMP (all upregulated)
Arioka 2017 [191]	Inorganic phosphate	RAW-D cells	-	c-Fos, NFATc1, DC-STAMP
Tsukamoto 2016 [192]	Elcatonin	Mouse bone marrow cells	Mice tail suspension model (microgravity) treated with elcatonin	NFATc1, cathepsin K, ATP6v0d2
Dou 2016 [193]	Cyanidin (low dose promotes osteoclastogenesis while high dose inhibits it)	Mouse bone marrow monocytes RAW264.7	-	c-Fos, NFATc1, Dual effects on Mitf, CD9, CD47, ATP6v0d2, DC-STAMP, OC-STAMP, OSCAR
Kuriya 2020 [194]	Tea extract (epigallocatechin gallate)	RAW264.7	-	DC-STAMP
Kanzaki 2017 [195]	PGDHC (promotes osteoclast multinucleation)	RAW264.7 Mouse bone marrow cells and peritoneal macrophages	Mice given calvarial injections of PGDHC	Myh9, Rac1, DC-STAMP (upregulated)
Ausk 2017 [195]	Botulinum toxin A causing muscle paralysis	Mouse bone marrow cells	Mice injected with botulinum toxin A in right calf muscle	TNF-α, IL-1, DC-STAMP, OC-STAMP
Fujita 2012 [197]	Vitamin E (α-tocopherol, increased osteoclast multinucleation)	Mouse bone marrow cells from Ttpa–/– and WT mice	WT mice treated with α-tocopherol	p38, Mitf, DC-STAMP (all upregulated)

LPS, lipopolysaccharide; MAPK, mitogen-activated protein kinase; NF-κB, nuclear factor-kappa B; NFATc1, nuclear factor of activated T-cells, cytoplasmic 1; Trap, tartrate-resistant acid phosphatase; DC-STAMP, dendritic cell specific transmembrane protein; Atp6v0d2, ATPase H+ transporting V0 subunit d2; OC-STAMP, osteoclast stimulatory transmembrane protein; OVX, ovariectomy; FAK, focal adhesion kinase; MFR, macrophage fusion receptor; OSCAR, osteoclast-associated receptor; Mitf, microphthalmia-associated transcription factor; PGDHC, phosphoglycerol dihydroceramide; Myh9, non-muscle myosin IIA; Rac1, RAS-related C3 botulinus toxin substrate 1; TNF-α, Tumor necrosis factor α; IL-1, interleukin-1; WT, wild type.

## 14. Conclusions

In this review, we have explored recent studies and summarized the molecules and signaling pathways involved in osteoclast multinucleation. Since the identification of the master fusogens DC-STAMP and OC-STAMP, our understanding of this process has progressed substantially in the 21st century. In addition, recently discovered molecules such as those on the siglec-15/DAP12 axis and the NETRIN-1/Flrt2/Unc5b axis as well as the purinergic receptors have been found to regulate osteoclast multinucleation as well. Despite the constant efforts of researchers to understand the mechanisms controlling osteoclast multinucleation from various angles, the process remains, to a large extent, unexplained. 

The process of osteoclast multinucleation is dynamic, complicated and finely controlled by multiple entangled factors. Multinucleation is considered to be the result of osteoclast fusion, but multiple fusion patterns have been observed between different fusion partners, depending on their cell morphology, i.e., whether they are mononuclear or multinuclear, and their stage of osteoclastogenesis [114]. Moreover, as Takegahara et al. have recently demonstrated, incomplete cytokinesis of osteoclasts can also cause multinucleation [3]; this finding offers us a completely new angle from which to investigate the process.

The fact that we have not yet identified the direct trigger for osteoclast multinucleation is hindering our understanding of the overall process. RANKL can induce osteoclastogenesis and subsequent multinucleation but is not the direct regulator of multinucleation. RANKL-unstimulated monocytes have been observed to fuse with RANKL-stimulated osteoclast progenitors [205], proving that RANKL signaling only indirectly induces osteoclast fusion. Deficiency of DC-STAMP and/or OC-STAMP completely abrogates osteoclast multinucleation but not osteoclast differentiation under RANKL stimulation, indicating that these two molecules act as “master fusogens” in osteoclasts, but their ligands are unknown. Researchers have developed innovative methods of directly triggering signals in osteoclast multinucleation; Verma et al., for example, have used LPC (lysophosphatidylcholine) to pause osteoclasts in a pre-fusion state in order to synchronize their fusion processes [115,206]. Chiu et al. engineered a DC-STAMP chimeric molecule in which light exposure mimics its ligation to explore its downstream signals [13]. The side effects of these interventions on osteoclasts remain unknown, however. There is an urgent need for the identification of the original physiological molecules that directly trigger osteoclast multinucleation.

In the last part of this review, we summarized recent preclinical studies that have focused on treatments targeting osteoclast multinucleation. At present, various drugs for the treatment of osteoporosis are under development, but many of the chemicals are pan-inhibitors of osteoclastogenesis, which means that they can also affect the activities of osteoblasts and osteocytes. Treatments that specifically target osteoclast multinucleation rather than osteoclast differentiation might attenuate this unexpected coupling and be worthy of more attention in the future. 

## Figures and Tables

**Figure 1 ijms-21-05685-f001:**
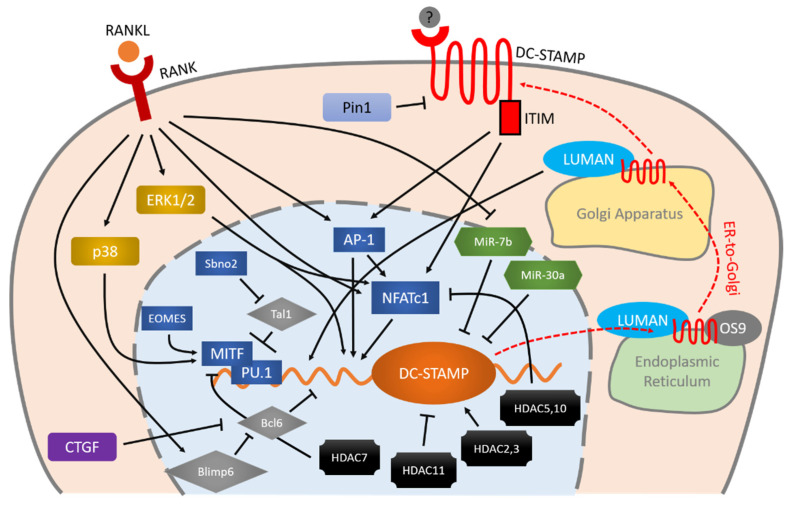
Schematic illustration of DC-STAMP regulation during osteoclastogenesis. RANKL-RANK is an important initiator of the expression of DC-STAMP via MAPK cascade, AP-1 and NFATc1. These two master transcription factors (AP-1 and NFATc1) and other factors including MITF, PU.1, Bcl6, and Blimp6 work together in a coordinated manner to control DC-STAMP with precision. Micro RNAs and HDACs also exhibit various effects on the regulation of DC-STAMP. Once the mRNA of *DC-STAMP* is translated into protein, it binds to LUMAN and OS9 for ER-to-Golgi transportation and eventually reaches the plasma membrane. Signaling through the ITIM that exists at the intracellular tail of DC-STAMP, conversely, regulates AP-1 and NFATc1 in a positive direction, further enhancing osteoclast differentiation and multinucleation. Arrows indicate positive regulation, bars indicate negative regulation, and red dashed arrows indicate intracellular transportation of DC-STAMP.

**Figure 2 ijms-21-05685-f002:**
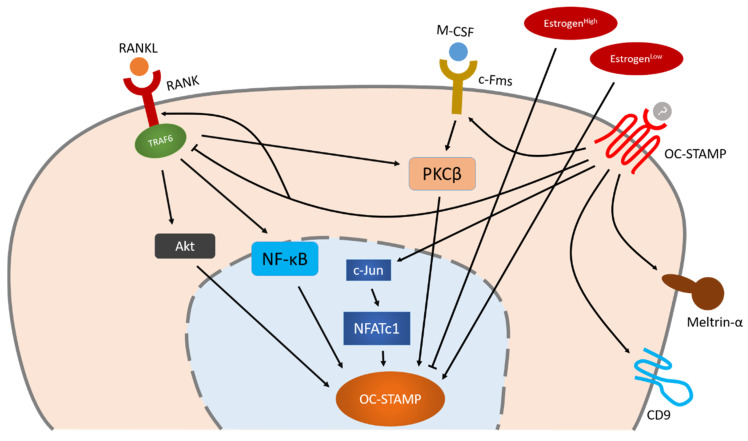
Schematic illustration of OC-STAMP regulation during osteoclastogenesis. NFATc1 is the master transcription factor of OC-STAMP, but Akt, NF-κB and PKCβ also play an important role in OC-STAMP’s regulation which is independent of NFATc1. OC-STAMP-induced signaling may positively influence other fusogens such as CD9 and meltrin-α, enabling them to regulate osteoclast multinucleation together. Arrow indicate positive regulation, bars indicate negative regulation.

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
