# Peer review of "Osteoclast Multinucleation: Review of Current Literature"

_ijms, 2020, doi:10.3390/ijms21165685_

Round 1

Reviewer 1 Report

The authors Kodama and Kaito reviewed in detail the current literature on multinucleation of osteoclasts. The manuscript is well written. A few suggestions to further improve the manuscript are as follow.

Title: May rephrase the title, ex., Osteoclast Multinucleation: Review of Current Literature” (or) “Review of current literature on osteoclast multinucleation”

P.3; line 110; clarify the sentence “…..chemokine ligand could be a potential ligand.”

P.4; Line 148; Subtitle 2.4. specify c-Fos as AP-1 is termed as a complex of c-Fos & Jun. Therefore, the authors may revise the subtitle ex., “c-Fos and NFATc1 modulation of DC-STAMP gene expression”

P.4; Line 176; Citation#40 Oh et al does not match in the bibliography list provided at the end. Please verify.

P.4; line 186; specify the subtitle for DC-STAMP- ex., “2.6. Bcl6 is a transcriptional repressor of DC-STAMP”

Revise the Figure 1. Title, ex., Schematic illustration of DC-STAMP regulation during osteoclastogenesis. They could simplify the illustration deleting the exogenous and endogenous MMPs and vitamin E labeling info as they have minor role and not detailed in the text.

P.8; Line 325; they may revise the subtitle with specifics ex., “3.3. Regulation of OC-STAMP expression during osteoclastogenesis”

P.9; line 368; may revise the title “Figure 2. Schematic illustration of OC-STAMP regulation during osteoclastogenesis.”

Reviewer 2 Report

This is a well organized and well written manuscript. This paper can be accepted for publication.

Novelty: This is a thorough review of osteoclast multinucleation. Novelty is high because there is no recent thorough review about this topic.

Significance of content: The review contains the thorough regulation mechanism of DC-STAMP and OC-STAMP as “master fusogens”. Besides, the authors also discuss about ATP6v0d2, CD47-MFR, cadherins, syncytins, ITAM-bearing immunoreceptors, NETRIN-1, Flrt2 and Unc5b axis, transglutaminases, purinergic receptors and TRAIL. The content is comprehensive.

Quality of presentation: The structure of this review is comprehensive. Two hundred and six references are up to date. The quality of presentation is good.

Scientific soundness: The authors reviewed all possible osteoclast nucleation regulation theories proposed now. Scientific soundness is good.

Interest to the readers: This review is very interesting for those who study bone metabolism. This is also a good review for scientists who newly want to approach bone metabolism.

Overall merit: After reading this review, the readers can have a good understanding of osteoclast multinucleation regulation.
